# Clinical, Biochemical, and Molecular Characterization of Two Families with Novel Mutations in the *LDHA* Gene (GSD XI)

**DOI:** 10.3390/genes13101835

**Published:** 2022-10-11

**Authors:** Pablo Serrano-Lorenzo, María Rabasa, Jesús Esteban, Irene Hidalgo Mayoral, Cristina Domínguez-González, Agustín Blanco-Echevarría, Rocío Garrido-Moraga, Alejandro Lucia, Alberto Blázquez, Juan C. Rubio, Carmen Palma-Milla, Joaquín Arenas, Miguel A. Martín

**Affiliations:** 1Mitochondrial and Neuromuscular Disorders Group, Hospital 12 de Octubre Health Research Institute (imas12), 28041 Madrid, Spain; 2Centro de Investigación Biomédica en Red de Enfermedades Raras (CIBERER), 28029 Madrid, Spain; 3Neurology Department, Hospital de Fuenlabrada, 28942 Madrid, Spain; 4Neuromuscular Unit, Department of Neurology, 12 de Octubre University Hospital, 28041 Madrid, Spain; 5Department of Internal Medicine, 12 de Octubre University Hospital, 28041 Madrid, Spain; 6Faculty of Sport Sciences, Universidad Europea de Madrid, 28670 Madrid, Spain; 7Department of Genetics, 12 de Octubre University Hospital, 28041 Madrid, Spain

**Keywords:** lactate dehydrogenase (LDH), lactate dehydrogenase A deficiency, *LDHA* gene, LDH isoenzymes, myopathy and psoriatic dermatitis

## Abstract

Lactate dehydrogenase (LDH) catalyzes the reversible conversion of L-lactate to pyruvate. LDH-A deficiency is an autosomal recessive disorder (glycogenosis type XI, OMIM#612933) caused by mutations in the *LDHA* gene. We present two young adult female patients presenting with intolerance to anaerobic exercise, episodes of rhabdomyolysis, and, in one of the patients, psoriasis-like dermatitis. We identified in the *LDHA* gene a homozygous c.410C>A substitution that predicts a p.Ser137Ter nonsense mutation in Patient One and a compound heterozygous c.410C>A (p.Ser137Ter) and c.750G>A (p.Trp250Ter) nonsense mutation in Patient Two. The pathogenicity of the variants was demonstrated by electrophoretic separation of LDH isoenzymes. Moreover, a flat lactate curve on the forearm exercise test, along with the clinical combination of myopathy and psoriatic-like dermatitis, can also lead to the diagnosis.

## 1. Introduction

### 1.1. LDH Enzyme

Lactate dehydrogenase (EC 1.1.1.27; L-lactate: NAD+ oxidoreductase; LDH) catalyzes the reversible conversion of L-lactate to pyruvate, with NAD+ as a hydrogen acceptor [1]. LDH plays a key role in anaerobic glycolysis as it regenerates nicotinamide adenine dinucleotide (NAD+) from NADH to provide ATP in situations where mitochondrial NADH oxidation cannot match the glycolysis rate, which can thus result in NAD+ depletion with a subsequent drop in glycolytic flux [2]. On the other hand, during anaerobic exercise, adenosine monophosphate (AMP) deaminase (AMPD, an enzyme encoded by the *AMPD1* gene) is activated in skeletal muscle. AMPD is an important regulator of muscle energy metabolism that is involved in the synthesis of inosine monophosphate (IMP), with the liberation of ammonia and displacement of the myokinase reaction toward ATP production [3].

LDH is composed of four peptide chains of two types, M (muscle, encoded by the *LDHA* gene) and H (heart, encoded by the *LDHB* gene), with their different combinations forming five tetrameric isoenzymes: LDH-1 (H4), LDH-2 (H3M), LDH-3 (H2M2), LDH-4 (HM3), and LDH-5 (M4)—listed in order of decreasing anodal mobility [1,4]. LDH is widely found in the body, yet has tissue-specific expression. Thus, isoenzymes LDH-1 and LDH-2 predominate in the heart, kidneys, and erythrocytes, whereas LDH-4 and LDH-5 are mainly found in the liver and skeletal muscle [5]. In turn, LDH concentrations in most tissues are greater than those physiologically found in serum, and therefore, the leakage of the enzyme from damaged cells (e.g., in the event of haemolytic anemia or rhabdomyolysis) leads to remarkable increases in LDH activity in serum [1].

### 1.2. Lactate Dehydrogenase-A Deficiency

LDH-A deficiency (LDHA deficiency, glycogenosis storage disease [GSD] type XI; OMIM#612933) is an autosomal recessive disorder caused by pathogenic mutations in the *LDHA*, a gene that contains seven exons and spans approximately 12 kb in 11p15.1.

The clinical presentation of LDHA deficiency is characterized by exercise intolerance, with cramps, myalgia, and myoglobinuria due to rhabdomyolysis after strenuous exercise [6]. Myoglobinuria may lead to acute renal failure [7,8]. Furthermore, uterine pain and stiffness have been reported in pregnant patients [7], associated with an increase in serum pyruvate levels [9], and all labors were reported to require caesarean section.

Several skin lesions have been documented to be associated with the disease, such as desquamating erythematosquamous lesions, pustular psoriasis-like lesions, and annular erythematous plaques [10,11,12]. These conditions typically appear in spring and spontaneously resolve after the fall [11,12,13]. In some cases, LDHA-deficient patients only exhibit skin symptoms [13].

Since the first family with LDHA deficiency was described in 1980 [14], 14 different families with altered LDHA activity have been reported. Nearly all of them have been genetically characterized, except for a case in whom diagnosis was exclusively based on enzyme activity. Most of the reported mutations are deletions, with a 20 bp frameshift deletion being the most prevalent variant. One splicing mutation, three nonsense mutations, and one small insertion have also been described [6,7,8,10,12]. Although nearly all mutations have been characterized, the correlation between genotype and phenotype is still unclear. Affected cases with the same genotype show variable clinical manifestations and severity.

## 2. Materials and Methods

### 2.1. Case Report

#### 2.1.1. Family 1

A 17-year-old female showed myoglobinuria, myalgia, and pain throughout the body following an exercise stress test at school. She reported no weakness, but showed intolerance to exercise, especially to intense tasks of short-duration (i.e., ‘anaerobic’ exercise), as well as several episodes of rhabdomyolysis triggered by exertion. Basal serum creatine kinase (CK) values were increased (2-5X reference values, <170U/L). She had previously been diagnosed with psoriatic dermatitis (Figure 1). Laboratory tests displayed an abnormal response of the non-ischemic forearm exercise test: flat lactate curve (normal: lactate increase 4-6X) and exaggerated ammonium response with an increase of ≈30X (Figure 2) (normal: ammonium increase 5-10X). Both parents and the patient’s only sister were asymptomatic. Frequent mutations in the Caucasian population that are known to be associated with exercise intolerance (i.e., *PYGM* gene p.Arg50Ter, p.Gly205Ser, and p.Trp798Gln variants in the muscle isoform of glycogen phosphorylase, resulting in McArdle disease [GSDV] and the *CPT2* gene p.Ser113Leu variant in carnitine palmitoyltransferase) were absent.

#### 2.1.2. Family 2

An 18-year-old female was studied for hyper-CK-emia. In the past, she suffered from myalgia after exercising and medium-high intensity efforts resulted in the appearance of symptoms and dark urine in the following days. She experienced muscle weakness, exercise intolerance (after intense running), and subsequent episodes of rhabdomyolysis. Serum CK basal values were 2-5X greater than reference values (normal: <170 U/L). Laboratory tests displayed an abnormal response to the non-ischemic forearm exercise test: flat lactate curve (normal: lactate increase 4-6X) and exaggerated ammonium response, with an increase of ≈30X (Figure 2) (normal: ammonium increase 5-10X). Both parents were asymptomatic. She has no siblings. She showed no p.Arg50Ter, p.Gly205Ser, or p.Trp798Arg mutations in the *PYGM* gene nor the p.Ser113Leu mutation in the *CPT2* gene.

### 2.2. Molecular Genetic Studies

#### 2.2.1. Metabolic Myopathies Panel

Blood DNA was extracted using QIAamp DNA Blood Midi Kit (QIAGEN, Hilden, Germany). A customized next-generation sequencing (NGS) panel of 32 genes associated with inherited metabolic myopathies was designed using Ampliseq™, and sequencing was conducted using the PGM-Ion Torrent platform (Life Technologies, Carlsbad, CA, USA). Sequence alignment (ref. GRCh37/hg19) and variant detection were performed in Torrent Suite (TMAP-variantCaller plugin). The annotation and prioritization of variants were carried out by our own script integration with Annovar [15]. The theoretical coverage of this panel reaches 99.2%.

#### 2.2.2. Clinical and Methodological Validation

Variant prioritization was performed assuming an autosomal recessive or recessive X-linked inheritance using the following steps: (i) Identified variants with minor allele frequency < 0.05 in population database, including the 1000 Genomes Project (http://browser.1000genomes.org, accessed on 24 June 2022), Exome Variant Server (http://evs.gs.washington.edu/EVS, accessed on 24 June 2022), Genome Aggregation Database (gnomAD) (https://gnomad.broadinstitute.org, accessed on 24 June 2022), Single Nucleotide Polymorphism Database (dbSNP) (http://www.ncbi.nlm.nih.gov/snp, accessed on 24 June 2022), and Collaborative Spanish Variant Server (http://csvs.babelomics.org, accessed on 24 June 2022); (ii) Intronic variants localized far from 15 nucleotides of the exon/intron junction were discarded; (iii) Status and ranking of the variants were performed in the ClinVar database (http://www.ncbi.nlm.nih.gov/clinvar, accessed on 24 June 2022); (iv) Determined variant pathogenicity predictors; (v) Performed assessment of phylogenetic conservation.

#### 2.2.3. PCR Amplification and Sanger Sequencing

Family segregation studies and haplotype analyses were performed by Sanger sequencing. The *LDHA* gene, HPS5 biogenesis of lysosomal organelles complex 2 subunit 2 (*HPS5*), tumor susceptibility 101 (*TSG101*), and neuron navigator 2 (*NAV2*) genes (genes surrounding *LDHA* sequenced for haplotype analysis) were amplified from genomic DNA by conventional PCR. The sequences of the primers used to amplify the *LDHA* variants were:

5′-GTGTGAACGTTGAGCTTGGG-3′ and 5′-TGCAGTCAAAAGCCTCACCT-3′ for *LDHA* exon 4.

PCR products were purified using Illustra GFX PCR DNA and gel band purification kit (GE Healthcare, Chicago, IL, USA), followed by Sanger sequencing in a 3130xl Genetic Analyzer (Applied Biosystems, Warrington, UK) using a Big Dye Terminator Cycle Sequencing Kit (Applied Biosystems).

### 2.3. LDH Isoenzyme Determination

Plasma LDH isoenzymes (collected in a lithium heparin tube) were separated by electrophoresis in agarose gel using the SAS-1 LD Vis Kit in the SAS-1 Plus instrument (Helena Biosciences Europe, UK). This instrument uses a modification in the method, as described by Preston et al. [16]: activity bands are visualized with a colorimetric reagent using lactate as a substrate and NAD+ as an acceptor of electrons and staining using a tetrazolium salt. Electrophoresis was performed at 80 V and 15 °C for 20 min following incubation at 45 °C for 25 min with the reagents required for the identification of LDH isoenzymes.

### 2.4. Non-Ischemic Forearm Exercise Test

Lactate and ammonium levels were quantified in serum at basal conditions and after 1, 2, 3, 4, and 5 min, respectively, of the forearm exercise test [17].

## 3. Results

To identify the genetic causes of patient phenotypes, we performed a customized NGS panel including 32 genes associated with metabolic myopathies. In the index case of Family 1, we found 178 variants. Only one variant was prioritized after applying the established criteria and ruling out low-quality variants and sequencing artifacts: a homozygous c.410C>A substitution that predicts a p.Ser137Ter nonsense mutation in the *LDHA* gene (NM_005566). This variant was present in the gnomAD database with an allele frequency < 0.001% and was absent in the 1000 Genomes Project database. As the variant generates a premature stop codon (amino acid position 137 of 333), it is considered potentially pathogenic. Following the American College of Medical Genetics and Genomics (ACMG) guidelines for the interpretation of sequence variants [18], the variant was classified as pathogenic. This variant is not present in public databases of variants, such as ClinVar or dbSNP.

The presence of the homozygous variant c.410C>A (p.Ser137Ter) in the *LDHA* gene of the patient was confirmed by Sanger sequencing. The patient´s asymptomatic mother, father, and sister were analyzed for the variant, and were all found to be heterozygous.

In the index case of Family 2, we found 200 variants. After applying the established criteria and ruling out low-quality variants and sequencing artifacts, two variants in the *LDHA* gene (NM_005566) were prioritized: a heterozygous c.410C>A substitution (the same one found in homozygosity in the index case of Family 1) and a heterozygous c.750G>A (p.Trp250Ter) nonsense mutation. This variant was absent in gnomAD and the 1000 Genomes Project databases and was not present in public databases of variants, such as ClinVar or dbSNP. As the variant generates a premature stop codon (amino acid position 250 of 333), it is considered potentially pathogenic. Following the ACMG guidelines for the interpretation of sequence variants [18], this variant was classified as pathogenic.

The presence of the heterozygous mutations c.410C>A (p.Ser137Ter) and c.750G>A (p.Trp250Ter) in the *LDHA* gene in the patient was confirmed by Sanger sequencing. The patient’s mother was found to be heterozygous for the c.410C>A (p.Ser137Ter), with the c.750G>A (p.Trp250Ter) found in heterozygosity in the patient´s father, verifying that both variants were present in a biallelic form in the patient. Sequencing of three SNPs in the *HPS5* (rs76929453, NM_181507: c.1862+87C>T), *TSG101* (rs2658552, NM_006292: c.549-84G>A), and *NAV2* (rs12283929, NM_001244963: c.3972-49C>G) genes that surround the *LDHA* gene—the distance between these SNPs and the variant c.410C>A (p.Ser137Ter) was less than 100K nucleotides—showed the same haplotype for the mutant allele in both families (Figure 3).

In the index cases of both families, plasma LDH levels were within (or slightly above) the normal range, due to muscle damage (Table 1). Basal CK levels were also increased (2-5x the reference values in both cases). We performed electrophoretic separation and subsequent visualization of each LDH isoenzyme in the plasma of both patients. Compared with the normal five-band isoenzyme pattern in controls, the isoenzyme pattern of the patients revealed a complete absence of the isoenzymes containing the M subunit, showing only one band of the LDH-1 (H4) isoenzyme (Figure 4).

## 4. Discussion

The two patients presented here are the first described with LDHA deficiency (GSD XI) in the Spanish population. In the index case of the first family, we identified a novel homozygous nonsense variant, c.410C>A (p.Ser137Ter), in a young female patient with several episodes of rhabdomyolysis, mildly elevated hyper-CK-emia, intolerance to predominantly ‘anaerobic’ exercise (i.e., high-intensity exercise in a short time), myalgia, and psoriasis-like lesions. We also identified the c.410C>A (p.Ser137Ter) variant in a second family. The proband was compound heterozygous for another novel variant, c.750G>A (p.Trp250Ter), with a very similar clinical presentation to that of the index case from the first family, including poor tolerance to anaerobic exercise; however, the index case from the second family did not experience dermatitis.

As LDH is involved in anaerobic glycolysis, patients affected with LDHA deficiency are expected to have worse tolerance to anaerobic exercise than controls. The non-ischemic forearm exercise test in both patients reflected this deficit in anaerobic glycolysis through the lactate profile; both cases showed a flat curve, similar to the pattern reported in patients with McArdle disease (GSD V) [17]. The high increase in ammonium (Figure 2, ammonium increase of 25-30X, normal: 5-10X) can be explained as the result of the increased activity of AMPD, to compensate for the muscle energetic deficit during intense exertion in the context of LDH deficiency [3,19,20]. Marked elevations of venous pyruvate levels have also been described in patients with LDHA deficiency during the exercise test [14], although they were not measured in either of the patients presented here. Overall, the main difference in the lactate stress test results of patients with LDHA deficiency compared with those with the very common glycogenosis (i.e., GSD V) is the high response of pyruvate, as the lactate response is similar in both diseases [7,14].

Regarding biochemical results, the two patients reported here showed high serum basal CK concentrations. LDH basal levels were within reference values or slightly elevated in both cases (Table 1). The electrophoretic separation of the LDH isoenzymes showed a single band in both patients belonging to the LDH-1 or H4 isoform, demonstrating the total absence of all the isoforms in which the M subunit of the LDH enzyme participates. Furthermore, there was a marked increase in the intensity of the LDH-1 band in the patients compared with controls, compensating for the absence of the rest of the isoenzymes and partially explaining the normal (or even increased) plasma LDH levels found in both patients (Figure 4).

As noted, the two families shared the c.410C>A (p.Ser137Ter) variant, even though there was no known relationship between them. However, the index case of the first family and the mother of the second family (carrier of the shared variant) were from the same Spanish geographic region, suggesting a possible founder effect of this variant. By analyzing three SNP polymorphisms in three genes close to *LDHA* (distance < 100 Kbp), the same haplotype in the c.410C>A allele was observed in both families, so both families could be genetically related.

The two novel variants reported here generate a null allele, similar to most of the mutations in the *LDHA* gene published to date (three frameshifts, three nonsense, and one splicing). The only non-null variant published was one in-frame deletion of one amino acid (*LDHA*, NM_005566: c.372_374delTGT; p.Val125del) [9,21,22,23] (Table 2). Almost all of the reported patients (14 families to date) showed at least one of the three symptoms characteristic of LDHA deficiency (i.e., myopathy, rhabdomyolysis, or psoriasis-like dermatitis). Most patients with this condition have myopathy, 4/14 patients have recurrent rhabdomyolysis, and approximately half of the patients (8/14) have psoriasis-like dermatitis [6,7,8,10,12,13,14,21,24,25]. There is no genotype–phenotype correlation, as the most frequently reported mutation, a 20bp deletion of Japanese origin (c.759_778del20; p.Leu254Argfs*8), has been associated with all possible phenotypes: isolated muscle symptoms, isolated psoriasis, and both muscle symptoms and psoriasis [7,10,14,24,26].

## 5. Conclusions

We have identified two novel nonsense variants associated with LDHA deficiency in two young adult female patients presenting with exercise intolerance, rhabdomyolysis episodes, and psoriasis-like dermatitis (in one case). The pathogenicity of the variants has been demonstrated by the electrophoretic separation of LDH isoenzymes. At the same time, other tests, such as the lactate stress test and its characteristic profile of the pyruvate curve, as well as the clinical combination of myopathy and psoriatic-like dermatitis, may also guide the diagnosis.

## Figures and Tables

**Figure 1 genes-13-01835-f001:**
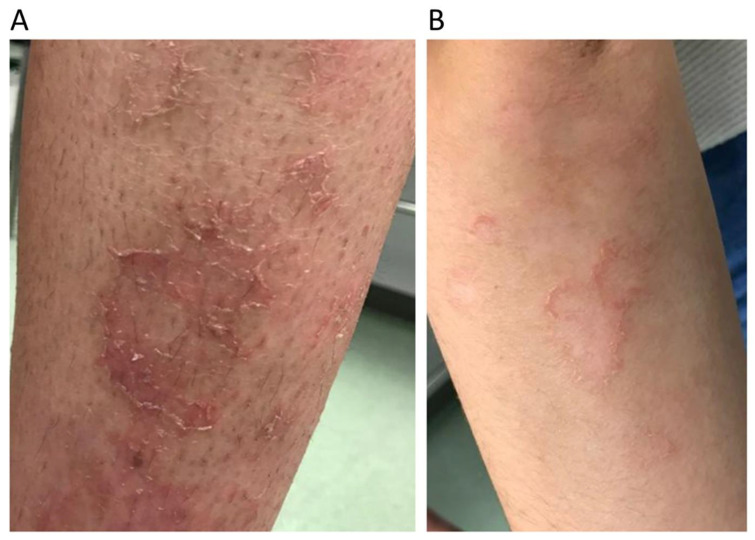
Psoriasis-like skin lesions in patient from Family 1 in lower (Panel **A**) and upper (Panel **B**) extremities.

**Figure 2 genes-13-01835-f002:**
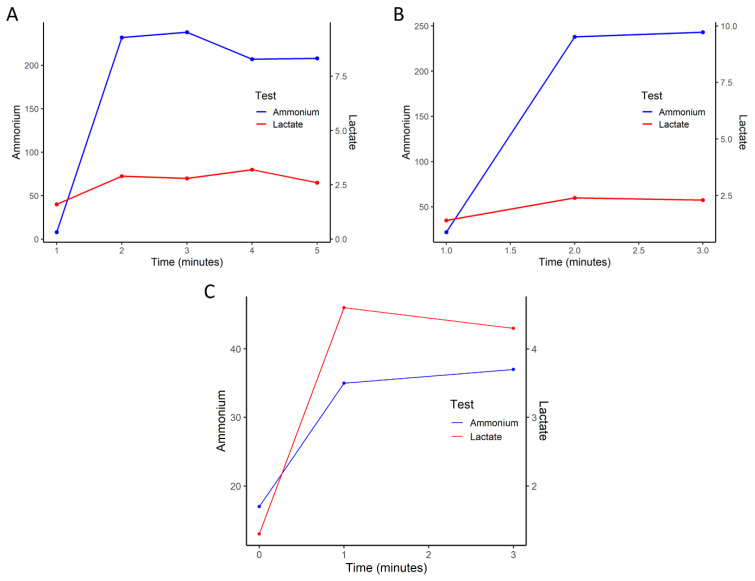
Non-ischemic forearm exercise test from index cases of Family 1 (Panel **A**) and Family 2 (Panel **B**), showing large (~30X) increases in ammonium concentration while lactate levels barely changed. Panel **C**: Example of a non-ischemic forearm exercise test.

**Figure 3 genes-13-01835-f003:**
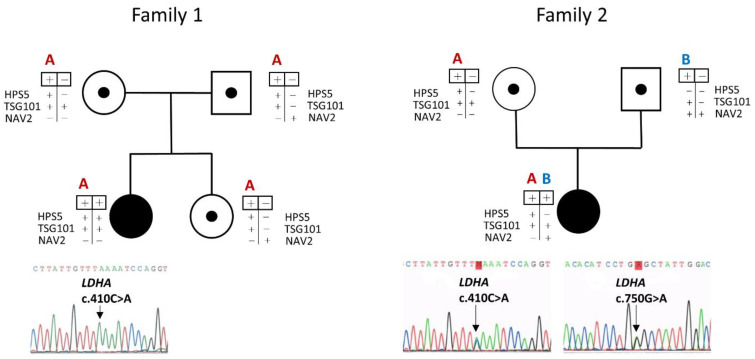
Family pedigrees illustrating both patient´s families representing the affected probands as black symbols. Rectangle −/−: wild type, +/− heterozygous, +/+ homozygous or compound heterozygote for the *LDHA* mutations (allele A: c.410C>A, allele B: c.750G>A). SNPs on *LDHA* neighboring genes *HPS5* (rs76929453, +: c.1862+87T, −: c.1862+87C), *TSG101* (rs2658552, +: c.549-84A, −: c.549-84G), and *NAV2* (rs12283929, +: c.3972-49G, −: c.3972-49C) for haplotype analysis are represented below the genotype of the *LDHA* mutations.

**Figure 4 genes-13-01835-f004:**
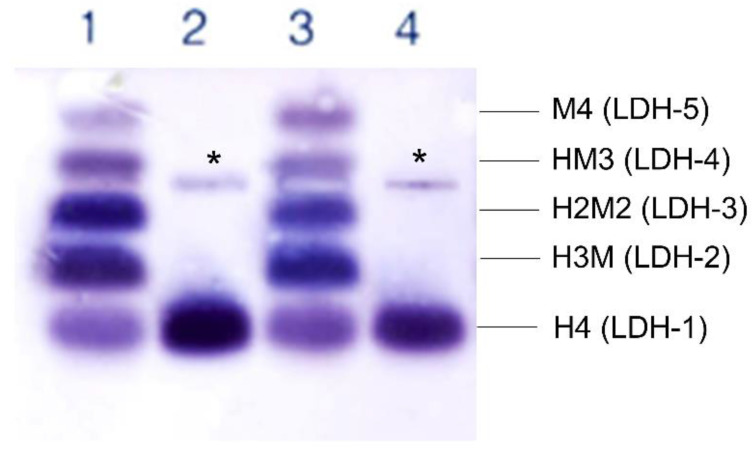
Electrophoretic separation of lactate dehydrogenase isoenzymes was obtained from plasma of two healthy controls (lanes 1 and 3), the index case of Family 1 (lane 2), and the index case of Family 2 (lane 4). Only one band of LDH-1 (H4) isoenzyme with the fastest mobility to the anodic side was present in both index cases. * mark caused by loading the sample.

**Table 1 genes-13-01835-t001:** Serum lactate dehydrogenase (LDH) and creatine kinase (CK) basal levels from index cases of both families.

Family	LDH (U/L)	CK (U/L)
1	191	315
2	232-247	491-846

Reference values: 1 LDH 135-214 U/L. 2 CK 34-145 U/L.

**Table 2 genes-13-01835-t002:** Lactate dehydrogenase A gene (*LDHA*) mutations and clinical symptoms.

Family	Age of Diagnosis	Age of Onset	Nucleotide Change	Amino Acid Change	Zygosity	Myopathy	Myoglobinuria	Dermatological Symptoms	Reference
1	18	10	c.759_778del20	p.Leu254Argfs*8	Homozygous	+	+	+	Kanno et al. [14]
2	26	5	c.759_778del20	p.Leu254Argfs*8	Homozygous	+	−	+	Maekawa et al. [24]
3	23	9	c.759_778del20	p.Leu254Argfs*8	Homozygous	+	+	−	Kanno et al. [7]
4	61	NA	c.759_778del20	p.Leu254Argfs*8	Homozygous	+	−	−	Maekawa et al. [26]
5	NA	NA	c.985G>T	p.Glu329Ter	Heterozygous (carrier)	−	−	−	Maekawa et al. [27]
6	51	Childhood	NA	NA	Homozygous	−	−	+	Nazzari et al. [13]
7	30	NA	c.640_641delCT	p.Leu214Alafs*7	Homozygous	+	+	+	Bryan et al. [8]
8	38	15	c.244G>A	IVS2 ds G-A -1	Homozygous	+	+	−	Tsujino et al. [6]
9	16	10	c.372_374delTGT	p.Val125del	Homozygous	+	−	+	Takayasu et al. [21]
10	55	Childhood	c.759_778del20	p.Leu254Argfs*8	Homozygous	−	−	+	Ito et al. [10]
11	34	Childhood	c.697C>T	p.Gln233Ter	Homozygous	+	−	+	Yue et al. [12]
12	54	NA	c.505C>T	p.Arg169Ter	Homozygous	NA	NA	+	Kanno et al. [25]
13	NA	NA	c.58dupC	p.Gln20Profs*3	Heterozygous (carrier)	NA	NA	NA	Kanno et al. [25]
14	NA	NA	c.759_778del20	p.Leu254Argfs*8	Homozygous	NA	NA	NA	Sudo [28]
15	17	17	c.410C>A	p.Ser137Ter	Homozygous	+	+	+	Present study
16	18	18	c.410C>A c.750G>A	p.Ser137Ter p.Trp250Ter	Compound heterozygous	+	+	−	Present study

NA: No data available.Regarding psoriatic lesions, some cases have described initial skin lesions that appeared at the beginning of the disease that later evolved into a more severe form of pustular psoriasis [10,23]. The decrease in NAD+ levels due to the LDHA deficiency may lead to alterations in calcium metabolism that could induce psoriatic lesions [23,29]. It has also been proposed that the ATP decrease in keratinocytes causes the release of proinflammatory factors, such as interleukin (IL)-8, vascular endothelial growth factor, or tumor necrosis factor α [10]. In the two cases presented here, only one showed dermatological symptoms, which were exacerbated after periods of stress (e.g., school exams). In most patients described [10,23] the season of the year also seems to influence symptom severity, so environmental factors may also play a role in the development of psoriasis-like lesions.

## Data Availability

The datasets and blots analyzed during the current study are available from the corresponding author upon reasonable request.

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
