# Peer review of "Clinical, Biochemical, and Molecular Characterization of Two Families with Novel Mutations in the LDHA Gene (GSD XI)"

_genes, 2022, doi:10.3390/genes13101835_

Round 1

Reviewer 1 Report

In this paper, the authors do explore the genotype of two families suspected of GSD XI by TGS in the Spanish population. This is an interesting, straight forward small (small sample size) clinical study on GSD XI. I felt the paper made good results.

However, the method section can be shorter.

Also, please add your identified mutation into your table so the reviewers/readers can investigate and compare them.

Author Response

Dear reviewer,

Many thanks for your comments. We are pleased to satisfy your suggestions and queries:

- The method section can be shorter:

We have shortened the Methods section by summarizing the Clinical and methodological validation and PCR Amplification and Sanger Sequencing subsections.

- Also, please add your identified mutation into your table so the reviewers/readers can investigate and compare them.

We have added our patient’s data to Table 2.

Reviewer 2 Report

Thank you for asking me to review this manuscript, which describes two families with members affected with glycogen storage disease type XI, providing clinical, biochemical and molecular genetic analyses and reporting two novel pathogenic variants in the LDHA gene. The clinical phenotype described is consistent with that known already in this rare disorder.

As such this is an interesting case report for a rare disorder although no new insights are given beyond the novel mutations.

Minor points: 

Line 52 - there is a typo where "LDH" is given as "LHD"

Line 94 - it would be better to give the genes before the amino acid/ nucleotide variants.

Line 110 - what gene do these variants relate to?

Line 181 - give reference for the "1000 genomes database" or explain what this is.

Figure 2: can you include normal response curves?

Table 2: Define "NA".

Author Response

Dear reviewer,

Many thanks for your comments. We are pleased to satisfy your suggestions and queries:

- Line 52 - there is a typo where "LDH" is given as "LHD".

We have corrected the typo.

- Line 94 - it would be better to give the genes before the amino acid/ nucleotide variants.

We change the order of the genes and mutations so now the gene names are first presented.

- Line 110 - what gene do these variants relate to?

We have included the gene names to which the variants refers.

- Line 181 - give reference for the "1000 genomes database" or explain what this is.

We have changed it to “1000 Genomes Project database” to use the same name referenced in the methods section (line 129).

- Figure 2: can you include normal response curves?

An example of a normal response curve have been added to Figure 2.

- Table 2: Define "NA".

Now NA is defined on the foot of Table 2.
